# Is multimodal occupational therapy in addition to usual care cost-effective in people with thumb carpometacarpal osteoarthritis? A cost-utility analysis of a randomised controlled trial

Anne Therese Tveter [iD],[1] Linn Kleven,[2] Nina Osteras,[1] Randi Nossum,[3] Ruth Else Mehl Eide,[4] Åse Klokkeide,[5] Karin Hoegh Matre,[4] Monika Olsen,[5] Ingvild Kjeken[1]

For numbered affiliations see end of article.

**Correspondence to**
Dr Anne Therese Tveter;
a.t.tveter@medisin.uio.no

## ABSTRACT

**Objective** The aim was to evaluate the cost-utility of a 3-month multimodal occupational therapy intervention in addition to usual care in patients with thumb carpometacarpal osteoarthritis (CMC1 OA).

**Methods** A cost-utility analysis was performed alongside a multicentre randomised controlled trial including three rheumatology departments in Norway. A total of 180 patients referred to surgical consultation due to CMC1 OA were randomised to either multimodal occupational therapy including patient education, hand exercises, assistive devices and orthoses (n=90), or usual care receiving only information on OA (n=90). The outcome measure was quality-adjusted life-years (QALYs) derived from the generic questionnaire EQ-5D-5L over a 2-year period. Resource use and health-related quality of life of the patients were prospectively collected at baseline, 4, 18 and 24 months. Costs were estimated by taking a healthcare and societal perspective. The results were expressed as incremental cost-effectiveness ratios, and a probabilistic sensitivity analysis with 1000 replications following intention-to-treat principle was done to account for uncertainty in the analysis.

**Results** During the 2-year follow-up period, patients receiving multimodal occupational therapy gained 0.06 more QALYs than patients receiving usual care. The mean (SD) direct costs were €3227 (3546) in the intervention group and €4378 (5487) in the usual care group, mean difference €−1151 (95% CI −2564, 262). The intervention was the dominant treatment with a probability of 94.5% being cost-effective given the willingness-to-pay threshold of €27 500.

**Conclusions** The within-trial analysis demonstrated that the multimodal occupational therapy in addition to usual care was cost-effective at 2 years in patients with CMC1 OA.

**Trial registration number** NCT01794754.

## STRENGTHS AND LIMITATIONS OF THIS STUDY

⇒ A strength of the study is the involvement of a patient research partner throughout the conduct of the study, although it may be considered a limitation that only one partner was involved.
⇒ Self-reported costs combined with several months between assessments may have biased the total costs of the study.
⇒ We were unable to include medication costs, travel expenses or purchase of technical or medical equipment (except those provided as part of the study) in the analyses due to imprecise reporting.

carpometacarpal osteoarthritis (CMC1 OA) affecting the base of the thumb.[2] Approximately 22% of people aged ≥50 years have symptomatic CMC1 OA.[3] Compared with OA in the other finger joints, CMC1 OA is associated with more pain and functional limitations, negatively affecting work ability and quality of life.[2 4 5]

There is a common misconception that hand OA is a normal part of ageing, and that there is 'nothing to be done'.[6] Although there is no cure for OA, the European League Against Rheumatism (EULAR) recommends that all people with hand OA should be offered a multimodal intervention consisting of information, hand exercises and assistive devices as first-line treatment.[7] In addition, orthoses should be provided for patients with CMC1 OA.[7] Pharmacological therapy is recommended as a symptom relieving supplemental intervention, whereas surgery should only be considered when other treatments have failed to provide sufficient pain relief.[7] The multimodal intervention is shown to have a statistically significant beneficial

## INTRODUCTION

Hand osteoarthritis (OA) is the most common joint disease,[1] with the subcategory thumb

short-term effect on pain[8 9] and hand function[9] compared with or in addition to usual care. However, there is an evidence-to-practice gap regarding this first-line multimodal intervention, with only 20% reporting to have received this treatment before being referred to surgical consultation.[10] This intervention also presents a tendency towards reduction and delay in surgery rate over a 2-year period.[11] Although not significant, this tendency may still be of clinical importance if shown to be cost-effective.

Oppong *et al* performed a trial-based economic evaluation on joint protection only, hand exercises only and a combination of joint protection and hand exercises compared with leaflet and advice over a 12-month period in patients with hand OA.[12] They found hand exercises to be the most cost-effective alternative.[12] In the study by Adams *et al* however, providing orthoses only was found to be neither effective nor cost-effective.[13] Thus, the results regarding the different components of recommended treatment are not conclusive. To the best of our knowledge, no studies have assessed the cost-utility of the different components combined in a multimodal intervention following the EULAR recommendations for the treatment of hand OA or CMC1 OA.

The aim of this study was to perform a cost-utility analysis (CUA) of a 3-month multimodal occupational therapy intervention in addition to usual care versus usual care only in patients with CMC1 OA referred to surgical consultation. The outcome measure was quality-adjusted life-years (QALYs), and the costs were estimated both as direct and indirect costs and societal costs over a 2-year period taking a healthcare and societal perspective.

## METHODS
### Study design and setting
A CUA was performed alongside a multicentre randomised controlled trial (RCT) including rheumatology departments in three Norwegian hospitals (St. Olav's Hospital in Trondheim, Haukeland University Hospital in Bergen and Haugesund Rheumatism Hospital) and is described in the protocol article.[11 14] In Norway, most hospitals (including the three hospitals involved in this trial) are public and owned by the Norwegian government.

### Patient and public involvement
A patient research partner from the Patient Research Panel at Diakonhjemmet Hospital, who had lived with hand OA for several years, was involved throughout the project. She participated in project meetings where she gave input to the project plan, recruitment procedure and information material as well as feedback on the interpretation and dissemination of the results.

### Participants and randomisation
Patients referred by their general practitioner for surgical consultation due to CMC1 OA at the three rheumatology departments between 2013 and 2015 were considered eligible for the study if they could speak Norwegian and

had no cognitive deficits. Eligible patients received information about the study, a written consent form and a questionnaire by post. Patients who agreed to participate were scheduled for an appointment with an occupational therapist (OT) at the hospital within 2 weeks after referral. After a baseline assessment, the patients were randomised to either a 3-month multimodal occupational therapy intervention or usual care. In Norway, the waiting period between referral and the actual surgical consultation was at the time this study was conducted between 4 and 6 months, and the intervention was conducted during this waiting period. Patients were assessed again immediately before surgical consultation (approximately 4 months after baseline assessment), and 18 and 24 months after the baseline assessment (online supplemental figure A). The sample size of 180 participants was estimated based on an expected surgery rate of 70% during the 2-year follow-up, which was the primary endpoint of the RCT.[11 14]

### Intervention
During the waiting period between referral and surgical consultation, the intervention group received a multimodal occupational therapy intervention in line with recommended first-line treatment.[7] In short, at the baseline assessment, the patients received written and oral information on hand OA, ergonomic principals, and the use of assistive devices by an OT. They were instructed in a hand exercise programme to be performed at home three times per week for 12 weeks. They all received five common assistive devices (a bread knife and vegetable knife with built-up handles, an enlarged grip for opening bottles, a key for opening jars (the Jar Key), and a self-opening pair of scissors) for use when needed, and daytime and night-time orthoses that they were encouraged to use as much as possible.[14] The assistive devices were provided at no cost to the patients, while they had to pay a deductible for the orthoses. The patients in the intervention group consulted the OT two times during this intervention period: at the baseline assessment and again after 2 weeks to adjust the orthoses and exercise programme, implying that the intervention was based mostly on self-management. Patients referred for bilateral hand OA received treatment for both hands. At the follow-up assessments (4, 18 and 24 months), the patients in the intervention group could receive new orthoses, or the OTs could adjust the ones they already had.

The control group received usual care, which generally means staying on the waiting list for consultation in specialist healthcare, that is, receiving no treatment in specialist healthcare. For the purpose of this study, the control group were assessed at the hospital at baseline and all follow-up time points by an OT. They received general written and oral information about hand OA. The control group had the same possibility as the intervention group to ask questions and be given advice.

If necessary, patients in both groups could independently seek treatment in primary healthcare, such as a consultation with general practitioner or other healthcare

personnel (this information was collected as part of this study). Both groups had a surgical consultation with a medical specialist approximately 4 months after the baseline assessment. All patients paid a deductible for the assessment consultations (a maximum of €259 per year).

### Randomisation and blinding
Patients were randomised using a computer-generated list with a block size of 10, stratified by hospital. Envelopes were opened by the patients after receiving information on hand OA and completing baseline assessment. The group affiliation was known to both the patients and the OTs.

### Data collection
#### Health outcome
The health-related quality of life (HRQoL) was measured at baseline, 4, 18 and 24 months using the generic instrument EQ-5D-5L (EuroQol 5-dimension, www.euroqol.org).[15] The EQ-5D-5L is a preference weighted measure of HRQoL based on five dimensions: mobility, self-care, daily activities, pain/discomfort and anxiety/depression. For each dimensions the patient assesses five possible levels of problems, from no problems to severe.[15] QALYs were derived from the EQ-5D-5L utility score, using the preference score from a UK population. Since there is no Norwegian tariff, the recommendation in Norway is to use the British tariff.[16] QALYs are weighted for time (years of life) with an index that express the HRQoL (utilities) on a scale anchored at 0 (health state equivalent to death) and 1 QALY (full health).

#### Resource use and costs
Healthcare usage was collected from different sources at baseline, 4, 18 and 24 months. The OT reported the number of consultations related to the intervention at each assessment, whereas surgical procedures and postoperative treatment were collected from patients' medical record. In addition, the use of primary and specialist healthcare was self-reported by the patient. We have included the costs related to the OT assessment at baseline, 4 and 18 months for both groups (OT assessment at 24 months were not included as the usual care group then received the intervention). Although the OT assessment was done for study purposes, it was included in the cost estimates because we cannot rule out a possible positive effect of an appointment with an OT in specialist healthcare for the control group. We have not included medication costs, travelling expenses or purchase of technical or medical equipment (except assistive devices provided to patients in the intervention group) due to lack of or imprecise reporting. Many patients reported using medication when needed, making it impossible to calculate costs. Travelling expenses were not collected, and with regard to technical or medical equipment (except those provided as part of the project), it was not possible to determine if the patients had bought the equipment before or during the trial.

Data on sickness absence related to surgery were collected from patient medical records. The number of hours of informal help at home due to OA (from spouse, family or friends) and the productivity loss (due to any reason of sickness absence and disability benefits, excluding sickness absence related to surgery) were self-reported. If not explicitly indicated by the patient, we expected that the productivity loss remained the same between assessment points, thus, if the patient reported sickness absence of 50% at 4 months and 100% at 18 months, this was calculated as 50% productivity loss between 4 and 18 months, and 100% between 18 and 24 months.

The costs of specialist healthcare visits and surgery were calculated using diagnosis-related group (DRG) weights from 2019. DRG is a coding system used for administration of both clinical and financial activity in specialist healthcare. The DRG costs were estimated by multiplying the cost weighting for the specific DRG group of the patient with the unit cost of 2019.[17] Sickness absence and disability benefits were valued in the same way for all patients. Costs related to absence from work were estimated from official statistics on the average wage by sex and age group in 2019. The average daily productivity loss was valued as €336 including societal costs of 40% and was adjusted for the percentage of sickness absence/disability benefits.[18] The number of hours of informal help at home was valued at minimum wage of €18 per hour.[19] All costs are estimated in Euros, using the exchange rate 0.10 (€1=Norwegian kroner 10) from 2019.

Direct costs included costs related to specialist healthcare (medical specialist, OT, physical therapist, nurse, radiographs surgery, assistive devices) and primary healthcare (general practitioner, physical therapist, manual therapist, acupuncturist, blood samples). Indirect costs included productivity loss due to sickness absence (due to surgery or for other reasons) and costs related to help with chores at home.

### Statistics
Differences in QALYs between the two groups were estimated using the trapezoidal method (area under the curve).[20] To illustrate the statistical uncertainty surrounding the incremental cost-effectiveness ratio (ICER), a probabilistic sensitivity analysis with 1000 replications was performed using the bootstrap resampling method. The bootstrapped costs and effects can be illustrated on a cost-effectiveness plane (CE plane). The CE plane (figure 1) are divided into four quadrants; where a new treatment that is more effective and less costly is plotted in the southeast quadrant, while one more costly and less effective is plotted in the northwest quadrant. Figure 2 shows the cost-effectiveness acceptability curve, which presents the probability that the intervention is cost-effective in addition to usual care for hand OA. The willingness-to-pay threshold was set to €27 500 (Kr 275 000) due to the low severity of the condition.[21]

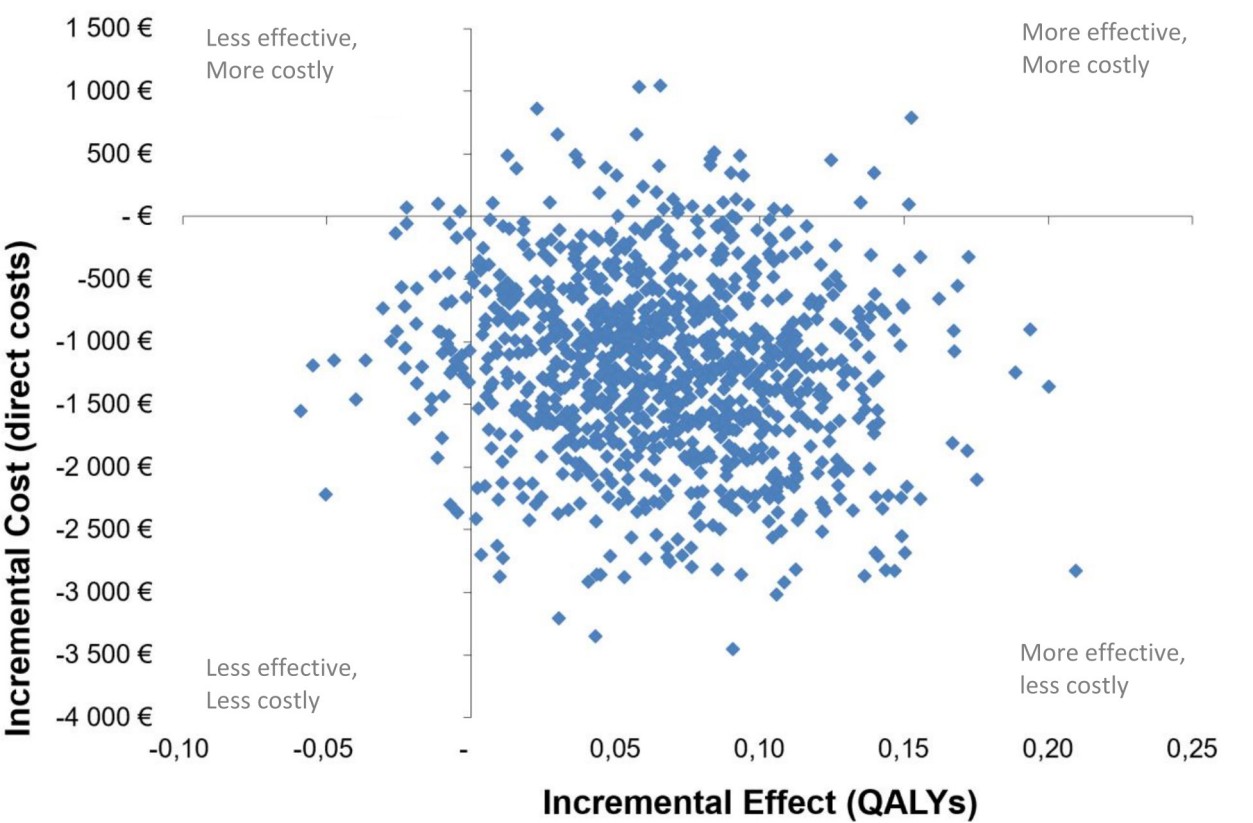

**Figure 1** Cost-effectiveness (CE) plane of the incremental cost-effectiveness ratios of multimodal occupational therapy (intervention group) in addition to usual care (control group) were recalculated with 1000 replications of the study data using the bootstrap resampling method. The intervention is at the origin. QALYs, quality-adjusted life-years.

Between-group differences in mean costs and mean effects were analysed using independent sample t-test. Significance level was set to 0.05. Statistical analyses were performed using Excel, SPSS V.26 and Stata V.16.

## RESULTS
### Baseline characteristics
A total of 180 patients were randomised to either multimodal occupational therapy in addition to usual care (n=90) or usual care only (n=90). The mean age across both groups was 63 years (SD 7.6) and 79% of the patients were women. We found no between-group differences in any of the baseline characteristics including utility score, medication use or percentage of patients working full-time or part-time (table 1).

### Missing data
A total of 18 patients (7 in the intervention group and 11 in the control group) had missing HRQoL utility score at one or more timepoints. For these missing values we used 'last observation carried forward'. Sensitivity analyses showed that the average HRQoL utility scores remained the same for both groups without imputation of these missing values. Missing self-reported costs were not imputed (n=16). The costs related to surgery were collected from patient medical records and were not encumbered by missing data.

### Cost-utility analysis
The total between-group difference in QALYs was 0.06 (95% CI −0.02, 0.15) after 24 months, in favour of the intervention group. The distribution of EQ-5D-5L utility score across the different time points for the two groups are shown in online supplemental figure B. The mean healthcare costs and productivity loss for both treatment groups are presented in table 2.

The mean cost of multimodal occupational therapy intervention (baseline information on hand OA, instruction in hand exercises, assistive devices, and customisation of orthoses, and adjustment of hand exercises and orthoses after 14 days) was approximately €500 per patient. Surgery accounted for the largest between-group difference related to direct healthcare costs with 22 surgical procedures in the intervention group and 33 surgical procedures in the control group, resulting in a €550 difference in surgical cost per patient between the two groups. The use of physiotherapy/occupational therapy also constituted a high cost. However, almost one-third of these costs were related to the predetermined assessment points (4, 18 and 24 months) that were part of the study but not the actual OT intervention (€375 per patient in the control group and €307 per patient in the intervention group).

Productivity loss due to sickness absence or disability benefits was the largest contributor to the total costs in

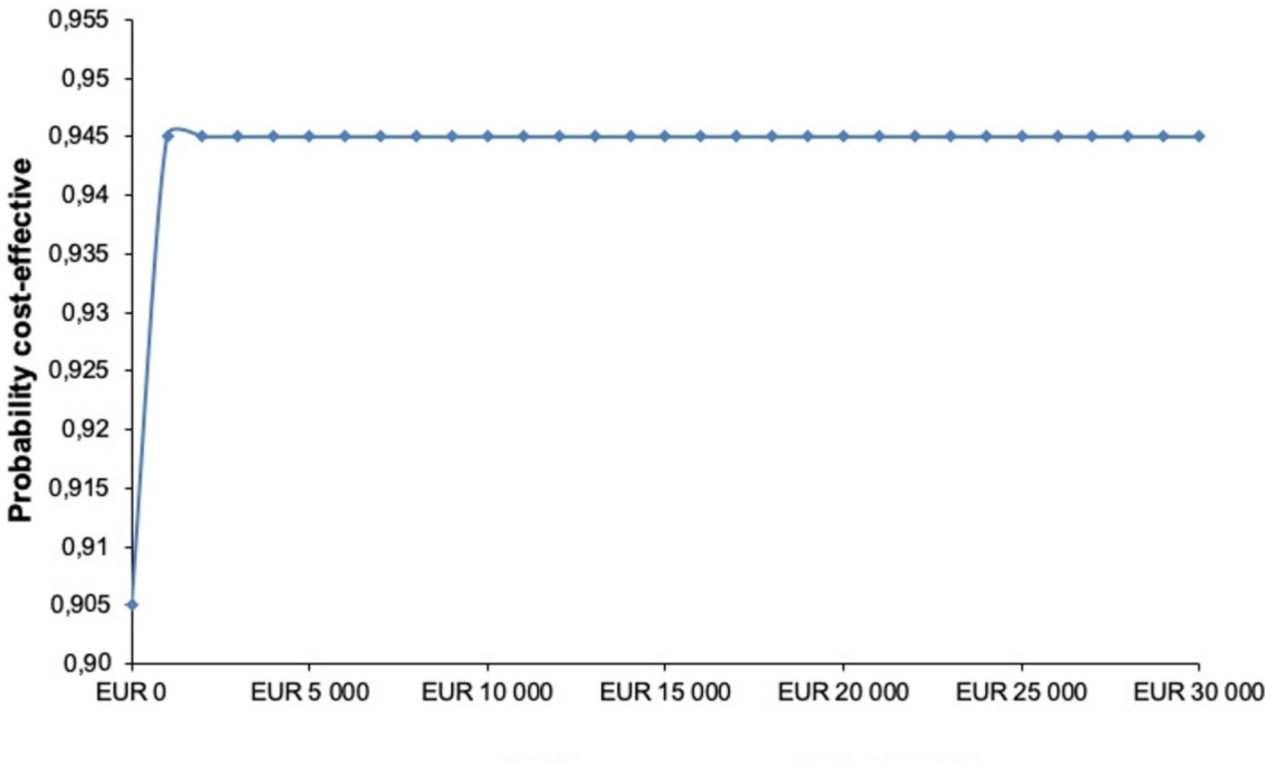

**Figure 2** Cost-effectiveness acceptability curve for direct costs, representing the probability that the multimodal intervention is cost-effective in addition to usual care. At a willingness-to-pay threshold of €27 500/QALY, the probability of multimodal intervention being cost-effective was 94.5%. QALYs, quality-adjusted life-years.

both groups. However, at baseline, only 7% of the sickness absence in both groups was reported to be due to hand OA, and the percentages at 4, 18 and 24 months were 6%, 2% and 0.5%, respectively. Similar numbers of patients in the two groups were retired at baseline (34% in the control group vs 32% in the intervention group), 4 months (36% vs 34%) and 24 months (42% vs 43%). However, more patients in the intervention group (43%) retired between 4 and 18 months than in the control group (36%), indicating that fewer patients contributed to productivity loss in the intervention group during this period.

Table 3 shows the mean incremental costs and incremental effects associated with direct and indirect healthcare costs. In the case of a negative ICER, the evaluated treatment is either dominant (more effective and less costly) or dominated (less effective and more costly). The intervention in this study was the dominant treatment (figure 1) with a 94.5% probability of being cost-effective given the willingness-to-pay threshold (figure 2).

## DISCUSSION

The aim of this study was to evaluate the cost-utility of a 3-month multimodal occupational therapy intervention in patients with CMC1 OA in a 2-year period. Our results demonstrate that the intervention, consisting of patient education, hand exercises, orthoses and assistive devices, has the highest probability of being cost-effective, given

the willingness-to-pay threshold (€27 500/QALY). The results indicate that the intervention is both less costly and more effective when provided alongside usual care. This study is one of few studies to evaluate the cost-utility of conservative treatment in hand OA, and the only study to evaluate the cost-utility of the 2018 EULAR guidelines on first-line treatment for CMC1 OA. Our findings add to the evidence base by demonstrating that multimodal occupational therapy is a cost-effective alternative for the management of CMC1 OA.

The results of this study are in line with a previous study showing that hand exercises were more cost-effective than leaflet and advice in patients with hand OA.[12] In contrast, the study by Oppong *et al* failed to show that the combination of hand exercises and use of assistive devices (joint protection classes[22]) were more cost-effective than hand exercises alone.[12] Similarly, the use of orthoses alone was shown to be neither effective nor cost-effective.[13] The results from the current study should, however, be replicated in new studies taking into account also travelling cost, medication, and technical and medical equipment.

In our study, we did not separately evaluate the different parts of the recommended first-line treatment. It is possible that we would have found the same results as Oppong *et al*,[12] given that hand exercises only entail costs related to the time used by health personnel to educate the patient, whereas both orthoses and assistive devices require an additional cost. On the other hand, the

**Table 1** Demographic variables for patients referred to surgical consultation due to thumb carpometacarpal osteoarthritis (n=180)

|  | Intervention group (n=90) | Control group (n=90) |
|---|---|---|
| Age, years, mean (SD) | 62.8 (7.5) | 63.3 (7.8) |
| Female sex, n (%) | 73 (81) | 69 (77) |
| Referred hand, n (%) |  |  |
| Right | 31 (34) | 17 (19) |
| Left | 24 (27) | 29 (32) |
| Both | 35 (39) | 44 (49) |
| Education level, n (%) |  |  |
| Primary school | 14 (16) | 20 (22) |
| High school | 44 (49) | 39 (43) |
| University/college ≤4 years | 17 (19) | 19 (21) |
| University/college >4 years | 15 (16) | 12 (13) |
| Living alone, n (%) | 17 (19) | 18 (20) |
| Working full/part-time, n (%) | 48 (53) | 43 (48) |
| Years with symptoms, median (IQR) | 5 (2–10) | 5 (2–10) |
| Presence of radiographic subluxation of the CMC1 joint | 58 (64) | 55 (61) |
| Comorbidity,* n (%) | 61 (68) | 56 (62) |
| Analgesics use (anti-inflammatory or pain relief medication), n (%) | 57 (63) | 57 (63) |
| Pain at rest (NRS 0–10, 0=no pain), median (IQR) | 3 (1–4) | 3 (2–5) |
| EQ-5D-5L utility score (1=best health), mean (SD) | 0.74 (0.16) | 0.73 (0.18) |

All information was collected at the baseline assessment.
Pain at rest was reported for referred hand (the mean of both hands if bilateral referral).
*Comorbidities was categorised as present if they had any of 16 predefined comorbidities.
CMC1, thumb carpometacarpal joint; NRS, Numeric Rating Scale.

combination of the recommended first-line treatment was found to have a significant 3-month short-term effect on reducing pain and improving hand function in these patients,[8] and should be offered to all patients with CMC1 OA as an effective and cost-effective alternative. Due to the self-managing nature of the intervention, the mean cost related to the multimodal occupational therapy intervention was low, approximately €500 per patient. However, the current study was conducted in specialist healthcare, and there are other options for delivering this intervention that may be even more cost-effective. The treatment of patients with OA should mainly be conducted in primary healthcare, possibly reducing the costs to some extent. However, due to the lack of availability of treatment options and long travel distances in parts of the country, more cost-effective options for delivery of treatment could be further explored, for example, using a smart phone application as shown in a Swedish study on digital first-line treatment for hip and knee OA.[23] We are currently evaluating the effect and cost-utility of a newly developed smartphone app (the Happy Hands app) which is designed to make recommended treatment available to people with hand OA.[24]

Surgery constituted a large cost in this within-trial. Although we found a tendency towards reduction in surgical procedures in the intervention group compared with the control group, this result was not statistically significant.[11] Still, the between-group difference in mean surgical costs (€550) was higher than the mean cost of the multimodal occupational therapy intervention alone (€500). Thus, although the difference in surgical rates between the two groups were found to be non-significant,[11] the difference may still be regarded as important in a healthcare and societal perspective as surgical procedures may be prone to more adverse events compared with multimodal occupational therapy.[11] Most likely, the avoidance of surgery has the highest impact on the cost-effectiveness of the intervention.

Productivity loss represented the largest contributor to the total costs in both groups. The between-group difference may to some extent be attributed to a higher retirement rate in the intervention group. Although most participants were of retirement age, we do not know the reason for retirement and this should therefore be reported in future studies. Additionally, to avoid bias, register data should be used for more accurate reporting of sickness absence and disability benefits.

The patients reported to have OA complaints for a median of 5 years, and more than 60% showed radiographic signs of subluxation of the CMC1 joint, indicating advanced disease progression. However, only 21% of the patients had visited a physiotherapist or an OT

**Table 2** Mean direct and indirect healthcare costs (mean and SD) per patient for the intervention and control group (n=180)

| | | Unit | Unit price, € | n | Intervention group Mean (SD) per patient, € | n | Control group Mean (SD) per patient, € | Mean difference (95% CI), € |
|---|---|---|---|---|---|---|---|---|
| Direct costs | Specialist healthcare | | | | | | | |
| | Medical specialist | Per visit | 205* | 136 | 309 (268) | 169 | 384 (365) | −75 (−169, 19) |
| | Occupational therapist including adaptation of orthoses | Per visit | 246* | 209 | 536 (318) | 65 | 158 (367) | 377 (275, 478) |
| | Occupational/physical therapist | Per visit | 205* | 193 | 428 (350) | 333 | 837 (866) | −355 (−549, −160) |
| | Nurse | Per visit | 205* | 1 | 16 (98) | 9 | 48 (189) | −32 (−76, 13) |
| | Inpatient surgery | Per surgery | 8 435* | 2 | 187 (1 250) | 6 | 562 (2 465) | −375 (−951, 201) |
| | Day-bed surgery | Per surgery | 2 251* | 20 | 500 (941) | 27 | 675 (1 190) | −175 (−491, 141) |
| | Radiographs | Per visit | 149¶ | 90 | 149 (0) | 90 | 149 (0) | – |
| | Assistive devices | Per person | 67¶ | 90 | 67 (0) | 0 | – | – |
| | Primary healthcare | | | | | | | |
| | General practitioner | Per visit | 32† | 91 | 32 (94) | 130 | 46 (104) | −14 (−43, 15) |
| | Physical therapist | Per visit | 31‡ | 187 | 64 (288) | 158 | 54 (160) | 10 (−59, 79) |
| | Manual therapist | Per visit | 52‡ | 0 | – | 16 | 10 (90) | – |
| | Acupuncturist | Per visit | 47¶ | 22 | 11 (64) | 44 | 23 (132) | −11 (−42, 19) |
| | Blood samples | Per test | 7† | 14 | 1 (6) | 24 | 2 (6) | −1 (−2, 1) |
| Indirect costs | Sickness absence due to surgery | Days | 336§ | 65 | 1212 (3815) | 64 | 1 199 (3817) | 186 (−1104, 1 141) |
| | Costs related to help with chores at home | Hours | 18§ | 3261 | 652 (1722) | 4612 | 922 (3089) | −270 (−1006, 465) |
| | Sickness absence and disability benefits (not sickness absence due to surgery) | Days | 336§ | 10351 | 38 299 (63 029) | 12 249 | 45 433 (66 320) | −7133 (−26 167, 1190) |

The table also shows unit price and the resource use (n).
All costs are given in Euro (€1=Kr 10) and adjusted to 2019.
*DRG (diagnosis-related group system).
†List of reimbursement codes.
‡Rates for physiotherapists.
§Official statistics of average wage.
¶Based on real costs.

before referral to surgical consultation.[10] Thus, referral to a multimodal occupational therapy intervention at an earlier stage in the disease course may be a possible step towards reducing the costs related to healthcare use.

The current study is the first study assessing the cost-utility of recommended multimodal treatment in patients with CMC1 OA. It is considered a strength that the study was conducted alongside an RCT. The study also has some limitations. Almost 80% of the sample were women, which represents a higher proportion of women than the usual gender distribution in hand OA. The results may therefore not be representative for men with hand OA.

**Table 3** Cost-utility analysis of mean costs (SD), mean effects (SD), incremental differences in costs and effects (95% CI) and incremental cost-effectiveness ratio (ICER) for direct and total direct and indirect costs (n=180)

| | Intervention group (n=90) | | Control group (n=90) | | Incremental cost (95% CI) | Incremental effect (95% CI) | ICER |
|---|---|---|---|---|---|---|---|
| | Mean cost (SD) | Mean effect* (SD) | Mean cost (SD) | Mean effect* (SD) | | | |
| Direct costs | 3227 (3 546) | 1.58 (0.28) | 4378 (5 487) | 1.51 (0.28) | −1151 (−2564, 262) | 0.06 (−0.02, 0.15) | Dominant |
| Total costs (direct and indirect costs) | 46 617 (65 499) | 1.58 (0.28) | 56 304 (68 872) | 1.51 (0.28) | −9688 (−9949, 29 324) | 0.06 (−0.02, 0.15) | Dominant |

Direct costs, €: self-reported use of primary and specialist healthcare, including surgery.
Indirect costs, €: productivity loss from sickness absence due to surgery, costs related to informal help at home, productivity loss from sickness absence and disability benefits excluding sickness absence due to surgery.
Total cost, €: direct costs+indirect costs.
*Mean effect without imputations; intervention group 1.58 (0.28), control group 1.51 (0.28).

The self-reported costs and the 14-month time frame between the reporting may have biased the total costs. This has most likely contributed to lower overall costs, as patients may not remember to report all consultations with healthcare personnel, especially in the time period between 4 and 18 months. On the other hand, the costs related to surgery were collected from patient medical records and will not be encumbered by the same bias. A total of 16 patients had missing data throughout the study. There may have been systematic reasons for these patients being missing, which could have affected the outcome. However, there were more missing data in the control group than the intervention group, indicating that the between-group difference could have been larger if all patients had reported direct and indirect costs. The results may have been biased by the method of imputation of missing HRQoL utility values. However, due to low number of missing values and sensitivity analyses showing no difference in mean score whether using 'Last observation carried forward' or not, we did not go any further with more advanced methods like multiple imputation. We were unable to include medication costs, travel expenses or purchase of technical or medical equipment, except for the assistive devices provided as part on the intervention, in the analyses. This is not in line with the most recent recommendations for cost-utility analyses[25] and may have biased the results by possibly underestimating the total costs. However, the equal distribution of medication use across the two groups imply that this do not constitute a large bias in the comparison of results. As with all conservative treatments, the patients knew that they received the intervention, and this may be a potential bias improving the effect measure. Measurement properties of the EQ-5D-5L has not been assessed in patients with CMC OA. The HRQoL measure might also capture QoL related to other conditions than only CMC OA. Although the follow-up sessions were similar for both groups, the results may have been influenced by the fact that patients in the intervention group could receive adjustment of orthoses or new orthoses at these sessions. This may have influenced the results. The involvement of a patient research partner with lived experience on hand OA is a strength, however, the study would have benefited from input from more than one partner. Future studies should evaluate the cost-utility of an even longer follow-up time as there is a lack of knowledge on surgical rate of these patients in the long term.

## CONCLUSION

The within-trial analysis demonstrated that multimodal occupational therapy was cost-effective in addition to usual care in patients with CMC1 OA, when taking a healthcare and societal perspective. The intervention was the dominant treatment with a 94.5% probability of being cost-effective at 2 years, given the willingness-to-pay threshold of €27 500/QALY. The low cost related to the intervention supports the use of this first-line treatment in all patients with CMC1 OA.

**Author affiliations**
[1]Center for Treatment in Rheumatic and Musculoskeletal Diseases (REMEDY), Norwegian National Advisory Unit on Rehabilitation in Rheumatology, Diakonhjemmet Hospital, Oslo, Norway
[2]Reviews and Health Technology Assessments, Norwegian Institute of Public Health, Oslo, Norway
[3]Department of Clinical services, St Olavs Hospital, Trondheim University Hospital, Trondheim, Norway
[4]Department of Rheumatology, Haukeland University Hospital, Bergen, Norway
[5]Department of Rheumatology, Haugesund Sanitary Association Rheumatism Hospital, Haugesund, Norway

**Acknowledgements** We would like to acknowledge patient research partner Øyvor Andreassen for her contribution throughout the project. We would also like to thank professor Tore K Kvien for reading and commenting on the manuscript. The project has been acknowledged at the EULAR Congress with the HPR Abstract Award in 2019 for the primary outcome results and the HPR Abstract Award in 2020 for the results presented in this article.

**Contributors** The authors have made substantial contributions to the concept or design of the work (IK, RN, REME, ÅK, KHM, MO, NO), the acquisition of data (RN, REME, ÅK, KHM, MO), analysis (IK, ATT, NO, LK) and interpretation of data (IK, ATT, NO, LK). All authors have contributed to the drafting and critical revision of the work (IK, RN, REME, ÅK, KHM, MO, NO, ATT, LK), final approval (IK, RN, REME, ÅK, KHM, MO, NO, ATT, LK) and agree to be accountable for all aspects of the work (IK, RN, REME, ÅK, KHM, MO, NO, ATT, LK). ATT is the guarantor, accepting full responsibility for the work, had access to the data, and controlled the decision to publish.

**Funding** This work was supported by the South-Eastern Norway Regional Health Authority (2015109) and Norges Forskningsråd (328657). The funding authorities had otherwise no involvement in the study. This research received no specific grant from any funding agency in the public, commercial or not-for-profit sector.

**Competing interests** None declared.

**Patient and public involvement** Patients and/or the public were involved in the design, or conduct, or reporting, or dissemination plans of this research. Refer to the Methods section for further details.

**Patient consent for publication** Not applicable.

**Ethics approval** This study involves human participants and was approved by Norwegian Regional Ethical Committee (2012/2265/REK sør-øst C). Participants gave informed consent to participate in the study before taking part.

**Provenance and peer review** Not commissioned; externally peer reviewed.

**Data availability statement** Data are available upon reasonable request. Data are available upon reasonable request to the project manager (Ingvild Kjeken, Ingvild. kjeken@diakonsyk.no).

**ORCID iD**
Anne Therese Tveter http://orcid.org/0000-0003-1701-9835

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
