## [Reviewer comments · BMJ Open]

ARTICLE DETAILS

TITLE (PROVISIONAL)	Is multimodal occupational therapy in addition to usual care cost-effective in people with thumb carpometacarpal osteoarthritis? A cost-utility analysis of a randomized controlled trial
AUTHORS	TVETER, Anne Therese; Kleven, Linn; Osteras, Nina; Nossum, Randi; Eide, Ruth Else Mehl; Klokkeide, Åse; Matre, Karin Hoegh; Olsen, Monika; Kjekken, Ingvild

VERSION 1 – REVIEW

REVIEWER	Buhler, Miranda University of Otago, School of Physiotherapy
REVIEW RETURNED	22-May-2022

GENERAL COMMENTS	Thank you for the opportunity to review this manuscript which reports the cost-utility analysis from an RCT investigating the effectiveness of adding multi-modal occupational therapy intervention to usual care of information, conducted 2013-2015. The study is well conducted and will make a valuable contribution to decision making by clinicians, health services, funders, and patients around non-surgical care provision for people with thumb CMC OA. I have some recommendations to improve the reporting of the study as follows: 1) I wonder if 'usual care' is an active comparator rather than no intervention comparator. If so, it would be more accurate to title and describe as 'effectiveness of multimodal intervention in addition to usual care' vs in comparison with, throughout the paper. My feeling is that information is an active intervention if it is thought to have some benefit which presumably it is. If you can give a compelling case for why your comparator is an inactive intervention, then just to note it should be 'comparison with' rather than 'comparison to'.2) Thank you for including the CHEERS checklist. You state that the study adopted a healthcare perspective. Please can you also include a statement about why this perspective was chosen? Thank you.3) You outline patient-public involvement to include the input of one individual. Would you be able to give further information about the demographics of this individual to better understand what context of the representative, and would you perhaps be able to add any limitations to this aspect in the Discussion? It also is not clear what is the 'study description' which they contributed to – is this design, reporting, recruitment information?4) Some details relating to the methods which should be made clearer are:a. P.6 Line 1: scheduled appointment within 'short period' – please state what the time frame was.
--

	b. 'The control group received usual care which usually means no treatment' – Please add the reference for this. c. In the methods, or perhaps the Introduction, please briefly contextualise the study in the health system conducted in, e.g. are primary and/or secondary services private or public? I note some information regarding waiting lists in referenced papers, but helpful to give brief context in this manuscript, particularly for purpose of contextualising the CE findings. d. Please give a clarifying statement about who was/was not blinded to group allocation. I see this information is in prior publications but should also be in this manuscript. The description of OT assessment is not clear regarding whether they are performing study data collection or care plan review, or both. 5) Results: Are you able to report any additional details about participant ethnicity and/or socioeconomic status of participants? 6) Discussion: P.15, given the health system context and study participant population, are there any comments you can make about whether digital health modalities might support or restrict access to care and outcomes for individuals or groups? 7) Discussion: P.16 Line 1: the meaning of this sentence is unclear. 8) Discussion: P.16 ... time of disease duration would be better reported as 'mean' (with SD). The final sentence of this paragraph does not quite flow logically on. Suggest rewording. 9) Discussion: P.16, major strength – I would disagree that 'large sample size' was study strength. I agree the sample size met the a-priori calculated target. However, I note that in the full study publication, post-hoc it was seen that the sample size may have been under-estimated, and the study not quite fully powered. Furthermore, >400 is a general ballpark figure for 'large' for an effectiveness study. The discussion point risks overinflating the confidence that can be had in the study finding. 10) For consideration: To what extent do you think the EQ5D captures impact on QoL for upper limb conditions such as CMC OA? If less sensitive, what impact on findings? I.e., may have been more influenced by other factors. This issue reflected in your finding that only a very small proportion of the sickness days attributable thumb CMC OA, and lost productivity the major cost. 11) For consideration: Given participant knowledge they are receiving the active intervention, and the active intervention involves devices which tend to have much greater nonspecific placebo effect, can you comment on the potential magnitude and direction of potential bias in study findings? 12) For consideration: two good quality RCTs investigating similar (but different) interventions in thumb CMC OA populations have recently been published (Adams et al., 2021; Deveza et al., 2021). Would you be able to make some comment about how your study findings compare with those of these two studies (I realise the primary outcomes differ, and only the one conducted a cost-effectiveness evaluation, yet it would still be worthwhile setting your study in the general landscape as this is likely to assist translation of findings overall into practice, and/or where future research should be directed, even if it is just to draw attention to [reported] differences in 'usual care' in different countries). Adams, J., Barratt, P., Rombach, I., Arden, N., Barbosa Boucas, S., Bradley, S., Doherty, M., Dutton, S. J., Goberman-Hill, R., Hislop-Lennie, K., Hutt-Greenyer, C., Jansen, V., Luengo-Fernandez, R., Williams, M., & Dziedzic, K. (2021). The clinical and cost effectiveness of splints for thumb base osteoarthritis: a randomized controlled clinical trial. Rheumatology (Oxford,
--	---

	England), 60(6), 2862-2877. https://doi.org/10.1093/rheumatology/keaa726 Deveza, L. A., Robbins, S. R., Duong, V., Bennell, K. L., Vicenzino, B., Hodges, P. W., Wajon, A., Jongs, R., Riordan, E. A., Fu, K., Oo, W. M., O'Connell, R. L., Eyles, J. P., & Hunter, D. J. (2021). Efficacy of a Combination of Conservative Therapies vs an Education Comparator on Clinical Outcomes in Thumb Base Osteoarthritis: A Randomized Clinical Trial. JAMA Intern Med, 181(4), 429-438. https://doi.org/10.1001/jamainternmed.2020.7101 13) Finally, the 'Strength and Weakness' section point 3 would benefit from being stated more clearly and informatively, e.g., 'self-reported costs and [up to 8 months – or what the time frame is] between evaluations may have biased total study costs [in the direction of... - how?]'.
--	---

REVIEWER	Sørensen, Jan The Royal College of Surgeons in Ireland, Healthcare Outcomes Research Center
REVIEW RETURNED	31-Oct-2022

GENERAL COMMENTS	This cost-utility analysis assesses the value of OT in patients with carpometacarpal OA. It is a clear and high-level description which is easy to follow. It is clear that the authors have been careful in following guidelines and standard practice. The results suggests that the intervention dominates the control intervention (usual practice). The resource use is only reported as costs in Euros. It is generally recommended to report both resource use and cost to enable readers to apply a different set of unit costs. I also note, that only accumulated cost data are reported and not for each data point. It might be possible to include these more detailed data as supplementary material. My main concern is lack of transparency in the outcome measures. The authors report 2-year QALYs 1.58 vs 1.51 although these are constructed by four self-reported data points. It would have been nice to see the score variation over time. The authors have chosen to report QALY as the only measure of effect. As a reader, I am wondering if the relative insensitive EQ is actually able to pick up a difference that can be attributed to the relative brief intervention. An improvement of 0.06 QALY is important. However, I really feel uncomfortable about this measure and would have liked to have seen some functional score changes and the correlation with the EQ-scores. I am not sure why the data in Table 2 and 3 are not aligned. It may be reporting og means and bootstrap medians, but the transparency in costs is lacking. There may need some commentary about this. The manuscript changes between using the term cost-utility and cost-effective. To me they mean and convey different things. CUA indicates use of a utility measure in reporting effects - and here the EQ-score is used as standard. The CUA aims to inform something about the value of this intervention in comparison with many other interventions and target groups - hence the application of a generic utility measure. The CEA indicates use of a different - non-
--

	utility effect measure - e.g. functional ability measure. This type of analysis aims to inform decision makers of the added value of using the intervention for the specified target group and inform decisions for this particular group of patients. I am aware that the American literature use CEA and does not differentiate between CEA and CUA - in contrast to the York/UK literature which is referred to here (Drummond et al). It might make the paper easier to read if a consistent term was used throughout the paper. In any event, I observe that CUA and CEA is used despite I think this is a CUA. This applies both in the introduction and especially in the discussion section. I am wondering if some sensitivity analysis would be relevant. I am concerned about the large QALY gain. May include report of gain with different assumptions relating to missing observations and perhaps score systems. A few smaller issues: I think the anonymity of sites is a bit artificial and generally not needed as I have full information about the locations of study through author identification and attached protocol paper. Remember to change these names in the final version :-) In the abstract, last sentence of method, the description of bootstrap is a bit strange. Does bootstrap really account for uncertainty? It is better described in the main text. Under results - could specify that the number in brackets after mean costs are standard deviation - or what it is. In main text under data collection - reference 14 does not seem appropriate here. That paper reports on the analysis of EQ-5D-3L scores. Ref 13 is probably ok to use - it is a bit dated and I actually thought there were Norwegian EQ-5D weights. The cross-walk scores is an interim solution. There may also be better UK EQ-5D-5L weight although I am aware that there were some challenges with the first version which was advised against by NICE. Replace xxx before tariff - Adjusting for time is probably wrong term - scores are weighted for time - which is better described somewhere else in the paper. Sentence in Resource use - under productivity - a bit unclear description of last observation carried forward. Is DRG really used for financial activity? Revise sentence.
--	---

REVIEWER	Holy, Chantal
	Johnson & Johnson Medical Devices, Real World Data Sciences
REVIEW RETURNED	31-Oct-2022

GENERAL COMMENTS	This study evaluated the cost-utility of a multimodal occupational therapy modality compared to no treatment, in patients with carpometacarpal osteoarthritis.
--

	This study was conducted alongside an RCT, and is therefore a prospective evaluation of randomized patients, a powerful study design. I would suggest the following, to strengthen the paper: 1) Detailed presentation of actual results. For all results (EQ-5D-5L and healthcare utilization), detailed descriptive statistics should be provided. For example, table 2 shows mean costs, but not the frequency of visits, and distribution thereof. If there is a cost difference, is it because a small number of patients in one group had a significantly larger number of visits than the average patients in the other group, or is it that all patients in the control group had slightly more care than in the experimental group? Providing detailed descriptive statistics on all the parameters (costs and units of utilization) that are used in the model will help readers understand why the therapy is cost-effective. 2) The discussion should include some assessment of why this therapy is cost-effective: is it simply that surgery was delayed? or are patients less likely to request care because they feel that they already got the information they needed? Or are they feeling better? The true impact of the therapy is unclear. 3) The EQ-5D-5L tool was used. The use of this tool for this particular diagnosis is not described in detail - how sensitive is it? has it been shown to have any ceiling or flooring effect for patients with CMC1 OA? can the authors expand a little on the possible limitation of using the EQ-5D-5L for this patient population? (It is also unclear: the EQ-5D-5L results shown on Table 1, are those baseline results (results at time of first visit?) Could a table be added to show the results at each follow-up time points?) Finally: there are a number of syntax issues throughout. For example, on page 14, line 54: "more cost-effective options could be to provide the intervention by utilizing a smart phone application" - this sentence is not incorrect, but could benefit from rephrasing. A review of the English syntax throughout would be helpful. (At many locations, the names of the hospitals are indicated with xxx - for example, page 5, lines 32, 41, page 6 line 6, page 27 line 7 etc...)
--	---

REVIEWER	Zomar, Bryn The University of British Columbia Faculty of Medicine, Orthopaedics
REVIEW RETURNED	04-Nov-2022

GENERAL COMMENTS	This is a well designed and performed economic analysis however there are a few minor revisions that should be addressed prior to publication.  - It's stated that the perspective for the analysis is healthcare (payer?), however as indirect costs are included this is really a societal perspective and should be acknowledged as such. - It's noted that only 10 subjects had missing data, however the supplemental flow diagram shows that 16 subjects missed at least one follow-up time point. Please provide further explanation for
--

	how the self-report data for those additional 6 subjects was dealt with.  - The published study protocol notes that the cost of medical or technical equipment purchased by subjects was collected. Why was this not reported in your results or included in your analyses? Please include these costs or include a statement as to why they were not. - Since medication use was recorded (as stated in your published protocol), why was this information not used in the calculation of the total costs? More explanation is needed. - Please add some additional background information about the healthcare system where the study takes place to provide context for the included costs (ie. is this a private or public-payer system). - Please present the EQ-5D-5L utility scores at each time point. Thank you for the opportunity to review this paper.
--	--

VERSION 1 – AUTHOR RESPONSE

Reviewer: 1

Dr. Miranda Buhler, University of Otago

Comments to the Author:

Thank you for the opportunity to review this manuscript which reports the cost-utility analysis from an RCT investigating the effectiveness of adding multi-modal occupational therapy intervention to usual care of information, conducted 2013-2015. The study is well conducted and will make a valuable contribution to decision making by clinicians, health services, funders, and patients around non-surgical care provision for people with thumb CMC OA. I have some recommendations to improve the reporting of the study as follows:

1)I wonder if 'usual care' is an active comparator rather than no intervention comparator. If so, it would be more accurate to title and describe as 'effectiveness of multimodal intervention in addition to usual care' vs in comparison with, throughout the paper. My feeling is that information is an active intervention if it is thought to have some benefit which presumably it is. If you can give a compelling case for why your comparator is an inactive intervention, then just to note it should be 'comparison with' rather than 'comparison to'.	We agree and have revised the manuscript accordingly	Compared to is changed to in addition to throughout the manuscript
2)Thank you for including the CHEERS checklist. You state that the study adopted a healthcare perspective. Please can you also include a statement about why this perspective was chosen? Thank you.	Thank you for this comment. Based on the comments from the other reviewers, we have included the societal perspective as well, as we have shown data on indirect costs as well.	Page 2, line 11: taking a healthcare and societal perspective Page 14, line 9: the difference may still be regarded as important in a healthcare and societal perspective as surgical procedures may be

		prone to more adverse events compared to multimodal occupational therapy.¹¹ Page 16, line 23: when taking a healthcare and societal perspective
3) You outline patient-public involvement to include the input of one individual. Would you be able to give further information about the demographics of this individual to better understand what context of the representative, and would you perhaps be able to add any limitations to this aspect in the Discussion? It also is not clear what is the 'study description' which they contributed to – is this design, reporting, recruitment information?	Thank you for this comment. We have revised the Patient and public involvement section	Page 4, lines 25-26, page 5, lines 1-2: A patient research partner from the Patient Research Panel at Diakonhjemmet Hospital with experienced knowledge on hand osteoarthritis was involved throughout the project. She participated in project meetings where she gave input to the project plan, recruitment procedure, and information material as well as feedback on the interpretation and dissemination of the results. Page 16, lines 16-18 The involvement of a patient research partner with lived experience on hand osteoarthritis is a strength, however, the study would have benefited from input from more than one partner

4)Some details relating to the methods which should be made clearer are: a.P.6 Line 1: scheduled appointment within ‘short period’ – please state what the time frame was.	Thank you for pointing this out. We have included the time frame	Page 5, line 9: within 2 weeks after referral
b.‘The control group received usual care which usually means no treatment’ – Please add the reference for this.	Thank you for this comment. Unfortunately, we do not have a reference on this, but we have elaborated on the description of what ‘no treatment’ means.	Page 6, lines 6-9 The control group received usual care, which generally means staying on the waiting list for consultation in specialist healthcare, i.e., receiving no treatment in specialist healthcare. They may however have been seeking treatment by themselves in primary healthcare (this information was collected as part of this study).
c.In the methods, or perhaps the Introduction, please briefly contextualise the study in the health system conducted in, e.g. are primary and/or secondary services private or public? I note some information regarding waiting lists in referenced papers, but helpful to give brief context in this manuscript, particularly for purpose of contextualising the CE findings.	Thank you for highlighting this. We have now included some information about the context in which this study was conducted.	Page 4, line 10-11 referred to surgical consultation in specialist healthcare Page 4, lines 19-21 In Norway, most hospitals (including the three hospitals involved in this trial) are public and owned by the Norwegian government.
d.Please give a clarifying statement about who was/was not blinded to group allocation. I see this information is in prior publications but should also be in this manuscript. The description of OT assessment is not clear regarding whether they are performing study data collection or care plan review, or both.	Thank you for this comment. We have now included information on randomization and blinding in the methods section.	Page 6, lines 18-21 Randomization and blinding Patients were randomized using a computer-generated list with a block size of 10, stratified by hospital. Envelopes were opened by the patients after receiving information on hand OA and completing baseline assessment. The group affiliation was known to both the patients and the OTs. Page 6, lines 14-16 The initial assessment and the 2-week follow-up in the intervention group

		were regarded as part of the care plan, while the subsequent follow-up assessments were performed as part of study data collection.
5)Results: Are you able to report any additional details about participant ethnicity and/or socioeconomic status of participants?	Thank you, but such information is rarely collected in Norway and was not collected in this study.	No action done
6)Discussion: P.15, given the health system context and study participant population, are there any comments you can make about whether digital health modalities might support or restrict access to care and outcomes for individuals or groups?	Thank you for highlighting this, we have included more information on this matter.	Page 14, lines 25-26, page 15, lines 1-3 However, due to the lack of availability of treatment options and long travel distances in parts of the country, more cost-effective options for delivery of treatment could be further explored, e.g. using a smart phone application as shown in a Swedish study on digital first-line treatment for hip and knee OA. ²¹ We are currently evaluating the effect and cost-utility of a newly developed smartphone app (the Happy Hands app) which is designed to make recommended treatment available to people with hand OA.
7)Discussion: P.16 Line 1: the meaning of this sentence is unclear.	Thank you for this comment. Unfortunately, we are unable to find the sentence as the page and line numbers you refer to do not seem to correspond with the proof that we got from the journal.	No action done.
8)Discussion: P.16 ... time of disease duration would be better reported as 'mean' (with SD). The	Thank you for this comment. Due to	Page 15, lines 12-13

final sentence of this paragraph does not quite flow logically on. Suggest rewording.	skewed data, we have reported disease duration as median and not mean. However, we have changed the wording from approximately to median. We have also included a sentence stating that median of 5 years disease duration and a large proportion with subluxation indicate advanced disease progression.	The patients reported to have OA complaints for a median of 5 years, and more than 60% showed radiographic signs of subluxation of the carpometacarpal joint, indicating advanced disease progression.
9)Discussion: P.16, major strength – I would disagree that 'large sample size' was study strength. I agree the sample size met the a-priori calculated target. However, I note that in the full study publication, post-hoc it was seen that the sample size may have been under-estimated, and the study not quite fully powered. Furthermore, >400 is a general ballpark figure for 'large' for an effectiveness study. The discussion point risks overinflating the confidence that can be had in the study finding.	Thank you for this feedback. We have removed that large sample size in itself was considered a strength.	Page 15, lines 17-19 The current study is the first study assessing the cost-utility of recommended multimodal treatment in patients with CMC1 OA. It is considered a strength that the study was conducted alongside a well-designed RCT.
10)For consideration: To what extent do you think the EQ5D captures impact on QoL for upper limb conditions such as CMC OA? If less sensitive, what impact on findings? I.e., may have been more influenced by other factors. This issue reflected in your finding that only a very small proportion of the sickness days attributable thumb CMC OA, and lost productivity the major cost.	Thank you for these questions. We have included information on the possibility limitations related to the HRQoL measure.	Page 16, lines 12-13 Measurement properties of the EQ-5D-5L has not been assessed in patients with CMC OA. The HRQoL measure might also capture QoL related to other conditions than only CMC OA.
11)For consideration: Given participant knowledge they are receiving the active intervention, and the active intervention involves devices which tend to have much greater nonspecific placebo effect, can you comment on the potential magnitude and direction of potential bias in study findings?	Thank you for this comment. We agree that the lack of blinding may be a possible bias, and have included this in the limitation section.	Page 16, lines10-12 As with all conservative treatments, the patients knew that they received the intervention, and this may be a potential bias improving the effect measure
12)For consideration: two good quality RCTs investigating similar (but different) interventions in thumb CMC OA populations have recently been published (Adams et al., 2021; Deveza et al., 2021). Would you be able to make some comment about how your study findings compare with those of these two studies (I realise the primary outcomes differ, and only the one conducted a cost-	Thank you for this suggestion. We have included the two studies in the introduction	Page 3, lines 23-24 The multimodal intervention is shown to have a significant short-term effect on pain^{8,9} and hand function⁹ compared to

effectiveness evaluation, yet it would still be worthwhile setting your study in the general landscape as this is likely to assist translation of findings overall into practice, and/or where future research should be directed, even if it is just to draw attention to [reported] differences in 'usual care' in different countries). Adams, J., Barratt, P., Rombach, I., Arden, N., Barbosa Boucas, S., Bradley, S., Doherty, M., Dutton, S. J., Gooberman-Hill, R., Hislop-Lennie, K., Hutt-Greenyer, C., Jansen, V., Luengo-Fernandez, R., Williams, M., & Dziedzic, K. (2021). The clinical and cost effectiveness of splints for thumb base osteoarthritis: a randomized controlled clinical trial. Rheumatology (Oxford, England), 60(6), 2862-2877. https://doi.org/10.1093/rheumatology/keaa726 Deveza, L. A., Robbins, S. R., Duong, V., Bennell, K. L., Vicenzino, B., Hodges, P. W., Wajon, A., Jongs, R., Riordan, E. A., Fu, K., Oo, W. M., O'Connell, R. L., Eyles, J. P., & Hunter, D. J. (2021). Efficacy of a Combination of Conservative Therapies vs an Education Comparator on Clinical Outcomes in Thumb Base Osteoarthritis: A Randomized Clinical Trial. JAMA Intern Med, 181(4), 429-438. https://doi.org/10.1001/jamainternmed.2020.7101		or in addition to usual care. Page 4, lines 6-7 In the study by Adams however, providing orthoses only was not found to be neither effective nor cost-effective.¹³ Page 14, lines 13-14 Similarly, the use of orthoses alone was not shown to be neither effective nor cost-effective.¹³
13) Finally, the 'Strength and Weakness' section point 3 would benefit from being stated more clearly and informatively, e.g., 'self-reported costs and [up to 8 months – or what the time frame is] between evaluations may have biased total study costs [in the direction of... - how?]'.	Thank you for this suggestion. We have now rephrased the sentence.	Page 15, line 21-22 The self-reported costs and the 14-month time frame between the reporting may have biased the total costs.

Reviewer: 2

Prof. Jan Sørensen, The Royal College of Surgeons in Ireland, Syddansk Universitet Comments to the Author:

This cost-utility analysis assesses the value of OT in patients with carpometacarpal OA.

It is a clear and high-level description which is easy to follow. It is clear that the authors have been careful in following guidelines and standard practice. The results suggests that the intervention dominates the control intervention (usual practice).	Thank you	
The resource use is only reported as costs in Euros. It is generally recommended to report both resource use and cost to enable readers to apply a different set of unit costs. I also note, that only accumulated cost data are reported and not for each data	Thank you for this feedback. We have now included the resource use (n) in Table 2	Table 2, page 11

point. It might be possible to include these more detailed data as supplementary material.		
My main concern is lack of transparency in the outcome measures. The authors report 2-year QALYs 1.58 vs 1.51 although these are constructed by four self-reported data points. It would have been nice to see the score variation over time.	We have included this in the supplementary file	Supplementary file B Page 10, lines 8-9 The distribution of EQ-5D-5L utility score across the different time points for the two groups are shown in Supplementary figure B. Page 19, line 6 Supplementary figure B EQ-5D-5L utility score across the different timepoints, divided by group.
The authors have chosen to report QALY as the only measure of effect. As a reader, I am wondering if the relative insensitive EQ is actually able to pick up a difference that can be attributed to the relative brief intervention. An improvement of 0.06 QALY is important. However, I really feel uncomfortable about this measure and would have liked to have seen some functional score changes and the correlation with the EQ-scores.	The trajectories of the scoring of pain and functional measures correspond to the trajectories of EQ5D utility score shown in supplementary file. We have however not included these results as they are currently being used in a submitted paper on pain and function.	No action done.
I am not sure why the data in Table 2 and 3 are not aligned. It may be reporting of means and bootstrap medians, but the transparency in costs is lacking. There may need some commentary about this.	Thank you for highlighting this. We understand the confusion. The numbers in Table 2 is the total costs pr group while the table 3 shows mean per group. We agree that these should align, however, to avoid duplicating the presented results we have removed these values from Table 2.	The total costs are removed from Table 2.
The manuscript changes between using the term cost-utility and cost-effective. To me they mean and convey different things. CUA indicates use of a utility measure in reporting effects - and here the EQ-score is used as standard.	Thank you for pointing this out. We had used CEA as an overarching term for both CEA and CUA, however, we have now tried to change this into CUA when talking about the analyses throughout the paper. In the description of the results	The wording is changed throughout the manuscript.

The CUA aims to inform something about the value of this intervention in comparison with many other interventions and target groups - hence the application of a generic utility measure. The CEA indicates use of a different - non-utility effect measure - e.g. functional ability measure. This type of analysis aims to inform decision makers of the added value of using the intervention for the specified target group and inform decisions for this particular group of patients. I am aware that the American literature use CEA and does not differentiate between CEA and CUA - in contrast to the York/UK literature which is referred to here (Drummond et al). It might make the paper easier to read if a consistent term was used throughout the paper. In any event, I observe that CUA and CEA is used despite I think this is a CUA. This applies both in the introduction and especially in the discussion section.	we will continue to use the wording cost-effective.	
I am wondering if some sensitivity analysis would be relevant. I am concerned about the large QALY gain. May include report of gain with different assumptions relating to missing observations and perhaps score systems.	Thank you for this question. Sensitivity analyses without missing observation show the same results as stated in the footnote in table 3. However, for transparency of the EQ5D utility measure, we have included the scoring over the different timepoints in a supplementary file.	Supplementary file B Page 10, lines 8-9 The distribution of EQ-5D-5L utility score across the different time points for the two groups are shown in Supplementary figure B. Page 19, line 6 Supplementary figure B EQ-5D-5L utility score across the different timepoints, divided by group.
I think the anonymity of sites is a bit artificial and generally not needed as I have full information about the locations of study through author identification and attached	Thank you, this is now changed throughout the paper	This is changed throughout the paper

protocol paper. Remember to change these names in the final version :-)		
In the abstract, last sentence of method, the description of bootstrap is a bit strange. Does bootstrap really account for uncertainty? It is better described in the main text.	Thank you for this comment. We have now changed the sentence to align more with the description in the main text.	Page 2, lines 12-13 and a probabilistic sensitivity analysis with 1 000 replications was done to account for uncertainty in the analysis.
Under results - could specify that the number in brackets after mean costs are standard deviation - or what it is.	Thank you for this comment. We have now stated that this is standard deviation	Table 2 and 3
In main text under data collection - reference 14 does not seem appropriate here. That paper reports on the analysis of EQ-5D-3L scores. Ref 13 is probably ok to use - it is a bit dated and I actually thought there were Norwegian EQ-5D weights. The cross-walk scores is an interrim solution. There may also be better UK EQ-5D-5L weight although I am aware that there were some challenges with the first version which was advised against by NICE.	Unfortunately, there are no Norwegian weights. Ref 14 is now removed	Reference removed.
Replace xxx before tariff -	Thank you, this is now changed	Page 7, line 3 Norwegian tariff
Adjusting for time is probably wrong term - scores are weighted for time - which is better described somewhere else in the paper.	Thank you. We have now changed the wording.	Page 7, line 4 QALYs are weighted for time
Sentence in Resource use - under productivity - a bit unclear description of last observation carried forward.	Thank you for pointing out the unclarity. We have now tried to describe this better.	Page 7, line 21-24 If not explicitly indicated by the patient, we expected that the productivity loss remained the same between assessment points, thus, if the patient reported sickness absence of 50% at 4 months and 100% at 18 months, this was calculated as 50% productivity loss between 4 and 18 months, and 100% between 18 and 24 months.
Is DRG really used for financial activity? Revise sentence.	Thank you for this question. In Norwegian hospitals, this is the financial system used	No action done

Reviewer: 3

Dr. Chantal Holy, Johnson & Johnson Medical Devices Comments to the Author:

This study evaluated the cost-utility of a multimodal occupational therapy modality compared to no treatment, in patients with carpometacarpal osteoarthritis.

This study was conducted alongside an RCT, and is therefore a prospective evaluation of randomized patients, a powerful study design.

I would suggest the following, to strengthen the paper:

1) Detailed presentation of actual results. For all results (EQ-5D-5L and healthcare utilization), detailed descriptive statistics should be provided. For example, table 2 shows mean costs, but not the frequency of visits, and distribution thereof. If there is a cost difference, is it because a small number of patients in one group had a significantly larger number of visits than the average patients in the other group, or is it that all patients in the control group had slightly more care than in the experimental group? Providing detailed descriptive statistics on all the parameters (costs and units of utilization) that are used in the model will help readers understand why the therapy is cost-effective.	Thank you for this comment. The frequency of visits is now included in table 2	Page 11, table 2 Frequency of resource use is included in Table 2 (n).
2) The discussion should include some assessment of why this therapy is cost-effective: is it simply that surgery was delayed? or are patients less likely to request care because they feel that they already got the information they needed? Or are they feeling better? The true impact of the therapy is unclear.	Thank you for this comment. We have tried to point out the most likely reason for why the therapy is cost-effective.	Page 15, lines 10-11 Most likely, the avoidance of surgery has the highest impact on the cost-effectiveness of the intervention.
3) The EQ-5D-5L tool was used. The use of this tool for this particular diagnosis is not described in detail - how sensitive is it? has it been shown to have any ceiling or flooring effect for patients with CMC1 OA? can the authors expand a little on the possible limitation of using the EQ-5D-5L for this patient population? (It is also unclear: the EQ-5D-5L results shown on Table 1, are those baseline results (results at time of first visit?) Could a table be added to show the results at each follow-up time points?)	Thank you for this comment. We have not found any measurement properties of the EQ-5D-5L in patients with hand OA / CMC OA. We have included a sentence regarding this in the limitation paragraph. A graph showing the EQ-5D-5L results across the 2 year follow-up is included in a supplementary file. Text has been added to Table 1 to show that this was information collected at baseline assessment	Page 16, lines 12-14 Measurement properties of the EQ-5D-5L has not been assessed in patients with CMC OA. The HRQoL measure might also capture QoL related to other conditions than only CMC OA. Supplementary file B Page 10, lines 8-9 The distribution of EQ-5D-5L utility score across the different time points for the two groups

		are shown in Supplementary figure B. Page 19, line 6 Supplementary figure B EQ-5D-5L utility score across the different timepoints, divided by group. Page 10, table 1 All information was collected at the baseline assessment
Finally: there are a number of syntax issues throughout. For example, on page 14, line 54: "more cost-effective options could be to provide the intervention by utilizing a smart phone application" - this sentence is not incorrect, but could benefit from rephrasing. A review of the English syntax throughout would be helpful.	Thank you. The sentence in the example has been rephrased. As we are not fluent in English, the manuscript has undergone professional proofreading. Unfortunately, we do not have the funding to send the manuscript for a second editing process.	Page 14, lines 25-26, page 15, lines 1-3 However, due to the lack of availability of treatment options and long travel distances in parts of the country, more cost-effective options for delivery of treatment could be further explored, e.g. using a smart phone application as shown in a Swedish study on digital first-line treatment for hip and knee OA.²³
(At many locations, the names of the hospitals are indicated with xxx - for example, page 5, lines 32, 41, page 6 line 6, page 27 line 7 etc...)	Thank you for this comment. This is changed throughout the manuscript	Changes have been made throughout the manuscript.

Reviewer: 4

Dr. Bryn Zomar, The University of British Columbia Faculty of Medicine, BC Children's Hospital

Comments to the Author:

This is a well designed and performed economic analysis however there are a few minor revisions that should be addressed prior to publication.

Thank you for the opportunity to review this paper.

- It's stated that the perspective for the analysis is healthcare (payer?), however as indirect costs are included this is really a societal perspective and should be acknowledged as such.	Thank you for this comment. We have included societal costs and have now included this perspective in the manuscript	Page 2, line 11: taking a healthcare and societal perspective Page 14, line 9: the difference may still be regarded as important in a healthcare and societal perspective as surgical procedures may be prone to more adverse events
---	---	--

		compared to multimodal occupational therapy.¹¹ Page 16, line 23: when taking a healthcare and societal perspective
- It's noted that only 10 subjects had missing data, however the supplemental flow diagram shows that 16 subjects missed at least one follow-up time point. Please provide further explanation for how the self-report data for those additional 6 subjects was dealt with.	Thank you for this, it was 10 patients who had missing data throughout the study (after baseline), the other 6 returned at some point. We have done the same for all patients and see that it is more correct to change from 10 to 16 in the manuscript.	Page 9, line 4 Missing self-reported costs were not imputed (n=16).
- The published study protocol notes that the cost of medical or technical equipment purchased by subjects was collected. Why was this not reported in your results or included in your analyses? Please include these costs or include a statement as to why they were not.	Thank you for highlighting this. Unfortunately, this reporting was difficult to use as we do not know if this was medical or technical equipment that they had already or that they had purchased as part of the study, or if they had any new equipment within the same category during the trial period. We have included a statement	Page 7. Lines 15-17 We have not included medication costs, travelling expenses or purchase of technical or medical equipment (except assistive devices provided to patients in the intervention group) due to lack of or imprecise reporting. Page 16, line 7 We were unable to include medication costs, travel expenses or purchase of technical or medical equipment in the analyses.
- Since medication use was recorded (as stated in your published protocol), why was this information not used in the calculation of the total costs? More explanation is needed.	Thank you for this comment. The reporting of medication use was very imprecise, mostly reported as "using pain medication when needed". We have now included a statement on why we did not include use this information	Page 7. Lines 15-17 We have not included medication costs, travelling expenses or purchase of technical or medical equipment (except assistive devices provided to patients in the intervention group) due to lack of or imprecise reporting.
- Please add some additional background information about the healthcare system where the study takes place to provide context for the included costs (ie. is this a private or public-payer system).	Thank you for pointing out the health care context. We have included a sentence about this in the manuscript.	Page 4, lines 19-21 In Norway, most hospitals (including the three hospitals involved in this trial) are public and owned by the Norwegian government.
- Please present the EQ-5D-5L utility scores at each time point.	Thank you. We have now included these results in a supplementary file.	Supplementary file B Page 10, lines 8-9

		The distribution of EQ-5D-5L utility score across the different time points for the two groups are shown in Supplementary figure B. Page 19, line 6 Supplementary figure B EQ-5D-5L utility score across the different timepoints, divided by group.
--	--	---

VERSION 2 – REVIEW

REVIEWER	Buhler, Miranda University of Otago, School of Physiotherapy
REVIEW RETURNED	21-Jan-2023

GENERAL COMMENTS	Thank you for the opportunity to re review this manuscript. It is a good study with interesting results. The manuscript is much improved with many points clarified. There are a few additional points to address and some suggestions on English language. To correctly convey the meaning that I think is intended in the Title, I suggest: Is multimodal occupational therapy in addition to usual care cost-effective in people with carpometacarpal osteoarthritis: A cost-utility analysis of a randomized controlled trial Abstract The intention to treat principle seems wrongly worded. Suggest: (QALYs) derived from the generic questionnaire EQ-5D-5L over a 2-year period. Resource use and health-related quality of life of the patients were prospectively collected at baseline, 4, 18, and 24 months. Costs were estimated by taking a healthcare and societal perspective. The results were expressed as incremental cost-effectiveness ratios (ICERs), and a probabilistic sensitivity analysis with 1 000 replications following intention-to-treat principle was done to account... Strengths  • I can't agree that the involvement of a single patient research partner was a strength. It is also a limitation. • ... due to lack of or imprecise reporting... just 'imprecise' would do Introduction Page 4 of 50  • Line 22: 'supplement' not the right term, suggest 'supplemental therapy' or supplemental intervention. • Line 23-24: 'positive' effect, or 'benefit'? Clarify 'significant' – statistical, clinically?
--

	 • Line 26: ‘This intervention’ – name the specific intervention... the ‘multimodal’? Page 5 of 50  • Lines 6-7: re write, avoid double negatives • Lines 3-8: re write to synthesis information better. What is the argument? Methods  • Lines 18-19: does care involve costs to patients? Or is it ‘free’ • Line 26: ‘experienced knowledge’ doesn’t make sense – she had (no. years) experience of... and / or was knowledgeable in... But I presume it’s her patient experience that you needed to have contribute to this study. Knowledgeable patients, while awesome, are not typical and might not bring the perspectives relevant to most other patients. Page 6 of 50  • Line 24: ‘they all received [at no cost, or at the cost of...{specify}]’ • Lines 6-9 and Lines 10-11: this is conflicting information. ‘the written and oral information received on hand OA’ is what constitutes, in this study, ‘usual care’, as far as your Methods lay it out. If this is not a standard part of the patient journey waiting on the surgical waitlist in Norway, then bring this up in the Discussion – i.e., what is ‘usual care’ in Norway for this patient group. But for the purpose of your research work, we need to know more about what this written and oral information is – who provided it (sounds like it might have been the specialist OT [Page 8 of 50, Line 15]?), were there opportunities to ask questions and advice, etc, so that we know what other active ingredients in the mix might also be. Recommend delete Lines 6-9 from methods. State what both groups received (we know it’s in the context of the study so no need to add that). State that the control [comparator] group received no other intervention. • Line 21: to have the participant so closely involved in the randomisation process is unusual. I would anticipate this may have added influence to their views on not getting the multimodal intervention. To what extent might the effect measured in this study be the true effect? And so how likely is it that the economic benefit found in this study would be realised in usual practice? Discuss in the Discussion. Page 8 of 50  • Lines 15-17: Please make a bit clearer what you had intended to do/tried to do, and what you actually did and why you changed. E.g., we set out to collect... data, but due to... data was missing or imprecise. *How did you know it was imprecise? I’d like to know more about what the problem was here, 1) to understand if there was any bias involved in not using the data and 2) to learn what needs solving for future work so it can be done better. I think any reader would like to know. • Line 25: ‘were calculated using DRG weights (diagnosis-related group)...’ change to, using diagnostic-related group (DRG) weights... Results  • Table 1: ‘Presence of subluxation of the CMC joint’ – how was this determined’ • Table 1: ‘Comorbidity’ – how was this defined? Some mention of the participant characteristics collected is needed in the Methods Page 13 of 50
--	---

	 • Line 8: why were the per patient costs for physio/OT associated with study assessments higher in the intervention vs control group? This seems odd. • Line 22: this is the interpretation of your findings and belongs your Discussion. Discussion Page 15 of 50  • Lines 13-14: revise to avoid double negatives • Line 18: give the definition of 'short-term' here Page 16 of 50  • Lines 4-6: doesn't make sense, re write. 'Even though...?' Page 17 of 50  • Lines 3-4: a missing word? • Lines 11-12: expand this discussion point. You haven't given us a clear sense of how confident we can be in your findings. Weigh up the potential magnitude of bias, and non-specific placebo effect. How could this be overcome in future study to give more confidence in the findings? • Include the timeframe in your concluding remarks. E.g., 95% probability of being cost-effective at 2 years... Add timeframe to Abstract conclusion as well.
--	---

REVIEWER	Holy, Chantal Johnson & Johnson Medical Devices, Real World Data Sciences
REVIEW RETURNED	29-Dec-2022

GENERAL COMMENTS	The authors have addressed all my comments. I do have a few additional observations and suggestions: 1) the syntax needs to be improved. For example: There are a couple of double negative sentences (on page 34 line 6 and on page 44: "was not found to be neither") - do the authors mean to say that it was found to be neither effective nor cost-effective (remove first "not")? On lines 10-11 page 34: surgical consultation in specialist healthcare: isn't all surgical consultation in specialist healthcare? can we remove the words "in specialist healthcare"? On line 7 page 36: suggest rewriting to: "they may, however, have been independently seeking primary care treatment". I am also not sure what the statement on page 36 line 14 adds: ("The initial assessment...") - could the authors please clarify: are some visits not accounted for in the cost analysis (and in Table 2) because they are "standard" and thus expected equally in both groups? Could this statement be clarified? 2) For Table 2: I don't understand the Occupational Therapy cost for the control group: with n = 5 and a unit cost of 246 euros, the table reports a cost per patient of 158 euros. Shouldn't this cost be much smaller (approximately in the range of $(5 \times 246) / 90 = 14$ euros)? Page 46: Please re-state that purchase of technical or medical equipment was not included EXCEPT for orthotic devices required for the intervention.
---

VERSION 2 – AUTHOR RESPONSE

Reviewer: 1		
Thank you for the opportunity to re review this manuscript. It is a good study with interesting results. The manuscript is much improved with many points clarified. There are a few additional points to address and some suggestions on English language. To correctly convey the meaning that I think is intended in the Title, I suggest: Is multimodal occupational therapy in addition to usual care cost-effective in people with carpometacarpal osteoarthritis: A cost-utility analysis of a randomized controlled trial	Thank you for this suggestion, this is now corrected	Page 1, title Is multimodal occupational therapy in addition to usual care cost-effective in people with carpometacarpal osteoarthritis? A cost-utility analysis of a randomized controlled trial
Abstract: The intention to treat principle seems wrongly worded. Suggest: (QALYs) derived from the generic questionnaire EQ-5D-5L over a 2-year period. Resource use and health-related quality of life of the patients were prospectively collected at baseline, 4, 18, and 24 months. Costs were estimated by taking a healthcare and societal perspective. The results were expressed as incremental cost-effectiveness ratios (ICERs), and a probabilistic sensitivity analysis with 1 000 replications following intention-to-treat principle was done to account...	Thank you for this suggestion, the wording is now changed	Page 2, lines 8-13: (QALYs) derived from the generic questionnaire EQ-5D-5L over a 2-year period. Resource use and health-related quality of life of the patients were prospectively collected at baseline, 4, 18, and 24 months. Costs were estimated by taking a healthcare and societal perspective. The results were expressed as incremental cost-effectiveness ratios (ICERs), and a probabilistic sensitivity analysis with 1 000 replications following intention-to-treat principle was done to account for uncertainty in the analysis
Strengths • I can't agree that the involvement of a single patient research partner was a strength. It is also a limitation.	Thank you, we have now included a sentence	Page 3, lines 5-6 • A strength of the study is the involvement of a patient research partner throughout the conduct of the study, although it may be considered

		a limitation that only one partner was involved
 ... due to lack of or imprecise reporting... just 'imprecise' would do 	Thank you, this is now changed	Page 3, lines 9-10: equipment in the analyses due to imprecise reporting.
Page 4 of 50  Line 22: 'supplement' not the right term, suggest 'supplemental therapy' or supplemental intervention 	Thank you for the suggestion, the wording is now changed	Page 3, line 21-22: Pharmacological therapy is recommended as a symptom relieving supplemental intervention,
 Line 23-24: 'positive' effect, or 'benefit'? Clarify 'significant' – statistical, clinically? 	Thank you for pointing this out. We have now included the wording statistically and beneficial	Page 3, lines 23-24 The multimodal intervention is shown to have a statistically significant beneficial short-term effect on pain ^{8,9} and hand function ⁹ compared to or in addition to usual care.
 Line 26: 'This intervention' – name the specific intervention... the 'multimodal'? 	Thank you, the word multimodal is now included	Page 3, line 25 there is an evidence-to-practice gap regarding this first-line multimodal intervention,
Page 5 of 50  Lines 6-7: re write, avoid double negatives 	Thank you for pointing this out. The word not is now removed.	Page 4, line 6: was found to be neither effective nor
 Lines 3-8: re write to synthesis information better. What is the argument? 	Thank you, we have now tried to argue better that studies assessing all the components combined in a multimodal intervention are lacking.	Page 4, lines 7-10: Thus, the results are regarding the different components of recommended treatment are not conclusive. To the best of our knowledge, no studies have assessed the cost-utility of the different components combined in a multimodal intervention following the EULAR recommendations for the treatment of hand OA or CMC1 OA.
Methods  Lines 18-19: does care involve costs to patients? Or is it 'free' 	Thank you for this comment. In Norway, the patients pay a deductible for all healthcare cost in primary and specialist with a maximum of € 259 per year. The deductible is	Page 6, lines 16-17: All patients paid a deductible for the assessment consultations (a maximum of €259 per year).

	included in the costs we have calculated.	
 Line 26: 'experienced knowledge' doesn't make sense – she had (no. years) experience of... and / or was knowledgeable in... But I presume it's her patient experience that you needed to have contribute to this study. Knowledgeable patients, while awesome, are not typical and might not bring the perspectives relevant to most other patients. 	Thank you, we agree and have changed the wording	Page 5, lines 1-2: , who had lived with hand osteoarthritis for several years, was involved throughout the project
Page 6 of 50  Line 24: 'they all received [at no cost, or at the cost of...{specify}] 	Thank you for this question. The assistive devices were provided at no cost for the participants.	Page 5, line 26: At no cost, they all received five common assistive devices...
 Lines 6-9 and Lines 10-11: this is conflicting information. 'the written and oral information received on hand OA' is what constitutes, in this study, 'usual care', as far as your Methods lay it out. If this is not a standard part of the patient journey waiting on the surgical waitlist in Norway, then bring this up in the Discussion – i.e., what is 'usual care' in Norway for this patient group. But for the purpose of your research work, we need to know more about what this written and oral information is – who provided it (sounds like it might have been the specialist OT [Page 8 of 50, Line 15]?), were there opportunities to ask questions and advice, etc, so that we know what other active ingredients in the mix might also be. Recommend delete Lines 6-9 from methods. State what both groups received (we know it's in the context of the study so no need to add that). State that the control [comparator] group received no other intervention. 	We appreciate this comment. We have revised the section regarding the intervention.	Page 5 and 6: See section about the Intervention.
 Line 21: to have the participant so closely involved in the randomisation process is unusual. I would anticipate this 	Thank you for this comment. This study was done in the waiting period between referral to consultation and the actual	No action done

may have added influence to their views on not getting the multimodal intervention. To what extent might the effect measured in this study be the true effect? And so how likely is it that the economic benefit found in this study would be realised in usual practice? Discuss in the Discussion.	consultation in specialist healthcare, thus, the patients were anticipating a consultation with an orthopedic surgeon, which they all got. We do not believe that the randomization process have influenced the result.	
Page 8 of 50  Lines 15-17: Please make a bit clearer what you had intended to do/tried to do, and what you actually did and why you changed. E.g., we set out to collect... data, but due to... data was missing or imprecise. *How did you know it was imprecise? I'd like to know more about what the problem was here, 1) to understand if there was any bias involved in not using the data and 2) to learn what needs solving for future work so it can be done better. I think any reader would like to know. 	Thank you for this comment. We have now added more information about why these costs were not included.	Page 7, lines 18-21 Many patients reported using medication when needed, making it impossible to calculate costs. Travelling expenses were not collected, and regarding technical or medical equipment, it was not possible to determine if the patients had bought the equipment before or during the trial.
 Line 25: 'were calculated using DRG weights (diagnosis-related group)...' change to, using diagnostic-related group (DRG) weights... 	We appreciate this suggestion. We have changed accordingly	Page 8, lines 3-4: using diagnosis-related group (DRG) weights
Results  Table 1: 'Presence of subluxation of the CMC joint' – how was this determined' 	Thank you for this question. Subluxation was determined based on radiographs. We have now included this in the table	Page 10, table 2: Presence of radiographic subluxation of the CMC joint
 Table 1: 'Comorbidity' – how was this defined? Some mention of the participant characteristics collected is needed in the Methods 	Thank you for this question. Comorbidity was categorized as being present if they had any of 16 predefined comorbidities	Page 10. Table 2: ‡ Comorbidities was categorized as present if they had any of 16 predefined comorbidities¹⁴.
Page 13 of 50  Line 8: why were the per patient costs for physio/OT associated with study assessments higher in the intervention vs control group? This seems odd. 	Thank you for this question. The intervention group received one assessment more (after 14 days) than the control group. Additionally, the cost for an assessment with an OT/physio is higher if an	No action done

	orthosis is adapted, and this was only done in the intervention group	
 Line 22: this is the interpretation of your findings and belongs your Discussion. 	Thank you, this line is now moved to the discussion.	Page 14, lines 5-6: given the willingness-to-pay threshold (€ 27 500/QALY), indicating that the intervention is both less costly and more effective in addition to usual care.
Page 15 of 50  Lines 13-14: revise to avoid double negatives 	We appreciate this suggestion. We have revised the sentence accordingly	Page 14, lines 14-15: Similarly, the use of orthoses alone was shown to be neither effective nor cost-effective
 Line 18: give the definition of 'short-term' here 	Thank you for this, we have included the timeframe in the sentence.	Page 14, line 22: was found to have a significant 3-month short-term effect on reducing pain and improving hand function in these patients
Page 16 of 50  Lines 4-6: doesn't make sense, re write. 'Even though...?' 	Thank you for pointing this out. We have changed the wording in the sentence	Page 15, line 8: Although we found a tendency towards reduction in surgical procedures in the intervention group compared to the control group
Page 17 of 50  Lines 3-4: a missing word? 	Thank you for noticing, we have revised the sentence	Page 16, line 7: The results may have been biased by the method of imputation of missing HRQoL utility values.
Discussion  Lines 11-12: expand this discussion point. You haven't given us a clear sense of how confident we can be in your findings. Weigh up the potential magnitude of bias, and non-specific placebo effect. How could this be overcome in future study to give more confidence in the findings? 	Thank you for this comment. We have now added a sentence highlighting what should be done in future studies.	Page 14, line 15-17: The results from the current study should, however, be replicated in new studies taking into account also travelling cost, medication, and technical and medical equipment.

 Include the timeframe in your concluding remarks. E.g., 95% probability of being cost-effective at 2 years... Add timeframe to Abstract conclusion as well. 	Thank you. We have now included the timeframe in the abstract and conclusion.	Page 2, line 20: The within-trial analysis demonstrated that the multimodal occupational therapy in addition to usual care was cost-effective at 2 years in patients with carpometacarpal osteoarthritis Page 17, line 3: treatment with a 94.5% probability of being cost-effective at 2 years
Reviewer: 3		
The authors have addressed all my comments. I do have a few additional observations and suggestions: 1) the syntax needs to be improved. For example: There are a couple of double negative sentences (on page 34 line 6 and on page 44: "was not found to be neither") - do the authors mean to say that it was found to be neither effective nor cost-effective (remove first "not")?	Thank you for this comment, we have revised the sentences with double negatives	Page 4, line 6: was found to be neither effective nor Page 14, lines 14-15: Similarly, the use of orthoses alone was shown to be neither effective nor cost-effective
On lines 10-11 page 34: surgical consultation in specialist healthcare: isn't all surgical consultation in specialist healthcare? can we remove the words "in specialist healthcare"?	Thank you, we have now removed specialist healthcare from the sentence	Page 2, line 5-6: A total of 180 patients referred to surgical consultation due to carpometacarpal OA were randomized Page 5, line 7: Patients referred by their general practitioner for surgical consultation due to CMC1 OA
On line 7 page 36: suggest rewriting to: "they may, however, have been independently seeking primary care treatment".	Thank you for this suggestion, we have revised the sentence accordingly	Page 6, line 13:

		They may however have been independently seeking treatment in primary healthcare
I am also not sure what the statement on page 36 line 14 adds: ("The initial assessment...") - could the authors please clarify: are some visits not accounted for in the cost analysis (and in Table 2) because they are "standard" and thus expected equally in both groups? Could this statement be clarified?	We appreciate the comment. Based on the comment from one of the other reviewers, we have revised the section about the intervention	Page 5 and 6: See section about the Intervention.
2) For Table 2: I don't understand the Occupational Therapy cost for the control group: with n = 5 and a unit cost of 246 euros, the table reports a cost per patient of 158 euros. Shouldn't this cost be much smaller (approximately in the range of $(5 \times 246) / 90 = 14$ euros)?	Thank you for noticing this. Unfortunately, a number was missing, this is now corrected	Page 10, Table 2
Page 46: Please re-state that purchase of technical or medical equipment was not included EXCEPT for orthotic devices required for the intervention...	Thank you for pointing this out. We have included your suggestion.	Page 16, lines 11-12: expenses or purchase of technical or medical equipment, except for the assistive devices provided as part on the intervention, in the analyses

VERSION 3 – REVIEW

REVIEWER	Buhler, Miranda University of Otago, School of Physiotherapy
REVIEW RETURNED	11-Apr-2023

GENERAL COMMENTS	Thank you to the authors for their responses, the manuscript is greatly improved as a result. Just one main concern remaining is the differential rate of retirement between groups and the potential confounding effect this may have had – see comments below. Otherwise, some minor suggestions as follows. HIGHLIGHTS A strength: Combined in a multimodal intervention, the EULAR recommendations for the treatment of hand OA are cost effective for CMC1 OA ABSTRACT
--

	'ICERs' abbreviation perhaps not needed as not repeated in the Abstract, although is usefully recognisable. Depending on journal style may wish to remove this. INTRODUCTION Carpometacarpal could refer to any CMC joint – suggest specifying in the title, abstract, and introduction. I note that CMC1 OA is used subsequently. I would suggest thumb carpometacarpal... abbreviated to CMC1 OA where appropriate. Pg.5 'Thus, the results are regarding the different components...' – 'are' should be deleted. METHODS Pg.7 '...patients in both groups could independently seeking treatment in...' - 'seek' rather than seeking. The 'Missing data' section would be better placed in the Results section. The Methods should outline what was planned to be done. RESULTS Pg.13 'Surgery accounted for the largest between-group difference...', however, Table 2 suggests that Sickness absence and disability benefits were the greatest between-group difference? Perhaps some clarification is needed. Is this paragraph related to health care costs? DISCUSSION The second sentence is long, recommend re writing. Pg.16 '...alongside a well-designed RCT.' How was the RCT 'well-designed'? Justify this statement. It seems that the differential rate of retirement between the two groups needs some discussion, particularly given that the lost productivity represented the greatest costs in both groups. It is possible that ending work earlier was a compromise and 'cost' that paradoxically reduced overall costs (in the intervention group) but might not represent a 'better' outcome. Is there a way this could be dealt with statistically, at least in future studies?
--	---

REVIEWER	Holy, Chantal Johnson & Johnson Medical Devices, Real World Data Sciences
REVIEW RETURNED	10-Apr-2023

GENERAL COMMENTS	The authors have addressed my comments. There are still syntax issues and one data inconsistency, to be resolved: 1) Data inconsistency: Table 2: Thank you for addressing the "OT including adaptation of orthosis" cost item. Another discrepancy is for O/PT cost: there are fewer events in the control group (152 vs 169) yet the cost is nearly double (837 vs 428), how is it possible? 2) Syntax issues: (There may be a few more, these are the ones I captured - suggest having a thorough review of all the syntax). Page 4 lines 7-8: remove the first "are" in the statement; "Thus, the results are regarding the different components of recommended treatment are not conclusive." Page 5, line 26: "At no cost, they all received" - at no cost to whom, to the patients? please specify. Page 6 line 13: "both groups could independently seek" (not seeking)
---

	Page 8 line 10: " help at home was valued" at minimum wage (instead of as a minimum wage) Table 1: the legend for comorbidity has a reference (reference 14) - this reference is for the study's protocol, but it doesn't give additional information on the 16 comorbidities, it says precisely what is stated here (comorbidity = presence of 16 predefined diseases). I would suggest removing the reference. Discussion: page 14 lines 5-6: suggest rewriting this statement as: "indicating that the intervention is both less costly and more effective when provided alongside usual care. "
--	--

VERSION 3 – AUTHOR RESPONSE

Reviewer 1:

Reviewer's comments	Authors' reply	Action
HIGHLIGHTS A strength: Combined in a multimodal intervention, the EULAR recommendations for the treatment of hand OA are cost effective for CMC1 OA	Thank you for this suggestion. However, in a previous revision the Editor stated that the Highlight section should contain bullet points that relate specifically to the methods. The novelty, aims, results or expected impact of the study should not be summarised here.	No action done
ABSTRACT 'ICERS' abbreviation perhaps not needed as not repeated in the Abstract, although is usefully recognisable. Depending on journal style may wish to remove this.	Thank you, we agree that this is not needed.	Page 2 , line 12: The results were expressed as incremental cost-effectiveness ratios,..
INTRODUCTION Carpometacarpal could refer to any CMC joint – suggest specifying in the title, abstract, and introduction. I note that CMC1 OA is used subsequently. I would suggest thumb carpometacarpal... abbreviated to CMC1 OA where appropriate.	Thank you for noticing. We have now changed the wording to thumb carpometacarpal osteoarthritis (CMC1 OA) throughout the title, abstract and introduction.	Page 1, title: Is multimodal occupational therapy in addition to usual care cost-effective in people with thumb carpometacarpal osteoarthritis? A cost-utility analysis of a randomized controlled trial Page 2, line 3: The aim was to evaluate the cost-utility of a 3-month multimodal occupational therapy intervention in addition to usual care in patients with thumb carpometacarpal osteoarthritis (CMC1 OA). Page 2, lines 6: A total of 180 patients referred to surgical consultation due to CMC1 OA were randomized...

		Page 2, line 20: The within-trial analysis demonstrated that the multimodal occupational therapy in addition to usual care was cost-effective at 2 years in patients with CMC1 OA. Page 3, line 10: with the subcategory thumb carpometacarpal osteoarthritis (CMC1 OA)
Pg.5 'Thus, the results are regarding the different components...' – 'are' should be deleted.		Page 4, line 5: Thus, the results regarding the different components of recommended treatment are not conclusive.
METHODS Pg.7 '...patients in both groups could independently seeking treatment in...' - 'seek' rather than seeking.		Page 6, line 12: If necessary, patients in both groups could independently seek treatment in primary healthcare,
The 'Missing data' section would be better placed in the Results section. The Methods should outline what was planned to be done.		Missing data section is moved to page 11, lines 1-6
RESULTS Pg.13 'Surgery accounted for the largest between-group difference...', however, Table 2 suggests that Sickness absence and disability benefits were the greatest between-group difference? Perhaps some clarification is needed. Is this paragraph related to health care costs?		Page 13, line 4: Surgery accounted for the largest between-group difference related to direct healthcare costs with 22 surgical procedures in the intervention group and 33 surgical procedures in the control group,
DISCUSSION The second sentence is long, recommend re writing.	Thank you for this suggestion. We have divided into two sentences.	Page 15, lines 5-6: The results indicate that the intervention is both less costly and more effective when provided alongside usual care.
Pg.16 '...alongside a well-designed RCT.' How was the RCT 'well-designed'? Justify this statement.	Thank you. We have moderated the sentence.	Page 17, lines 2-3: alongside an RCT
It seems that the differential rate of retirement between the two groups needs some discussion, particularly given that the lost productivity represented the greatest costs in both groups. It is possible	Thank you for highlighting this. We have included a paragraph where this is discussed.	Page 16, lines 16-20: Productivity loss represented the largest contributor to the total costs in both groups. The between-group difference may to some extent be attributed to a higher retirement rate in the

that ending work earlier was a compromise and 'cost' that paradoxically reduced overall costs (in the intervention group) but might not represent a 'better' outcome. Is there a way this could be dealt with statistically, at least in future studies?		intervention group. Although most participants were of retirement age, we do not know the reason for retirement and this should therefore be reported in future studies. Additionally, to avoid bias, register data should be used for more accurate reporting of sickness absence and disability benefits.
---	--	---

Reviewer: 3

1) Data inconsistency: Table 2: Thank you for addressing the "OT including adaptation of orthosis" cost item. Another discrepancy is for O/PT cost: there are fewer events in the control group (152 vs 169) yet the cost is nearly double (837 vs 428), how is it possible?	Thank you for addressing these issues. We have investigated the data, and while the costs are correct, the number of consultations per group for occupational therapist including adaptation of orthoses and occupational therapist/physical therapist in specialist healthcare are unfortunately wrong. This has now been corrected.	See Table 2. Intervention group: Occupational therapist including adaptation of orthoses: n=209 Occupational therapist/physical therapist: n=193 Control group: Occupational therapist/physical therapist: n=333
2) Syntax issues: (There may be a few more, these are the ones I captured - suggest having a thorough review of all the syntax). Page 4 lines 7-8: remove the first "are" in the statement; "Thus, the results are regarding the different components of recommended treatment are not conclusive."	Thank you for pointing out this error. The sentence is revised accordingly.	Page Thus, the results regarding the different components of recommended treatment are not conclusive.
Page 5, line 26: "At no cost, they all received" - at no cost to whom, to the patients? please specify.	Thank you for this comment. This was of no cost to the patients. This is now specified.	Page 5, line 27 and page 6, lines 1-2: The assistive devices were provided at no cost to the patients, while they had to pay a deductible for the orthoses.
Page 6 line 13: "both groups could independently seek" (not seeking)		Page 6, line 12: If necessary, patients in both groups could independently seek treatment in primary healthcare,
Page 8 line 10: " help at home was valued" at minimum wage (instead of as a minimum wage)		Page 8, line 8: The number of hours of informal help at home was valued at a minimum wage of € 18 per hour

Table 1: the legend for comorbidity has a reference (reference 14) - this reference is for the study's protocol, but it doesn't give additional information on the 16 comorbidities, it says precisely what is stated here (comorbidity = presence of 16 predefined diseases). I would suggest removing the reference.		Page 10, line 5: Reference removed
Discussion: page 14 lines 5-6: suggest rewriting this statement as: "indicating that the intervention is both less costly and more effective when provided alongside usual care." "		Page 15, line 5-6: The results indicate that the intervention is both less costly and more effective when provided alongside usual care.

VERSION 4 – REVIEW

REVIEWER	Buhler, Miranda University of Otago, School of Physiotherapy
REVIEW RETURNED	26-May-2023

GENERAL COMMENTS	This well conducted study makes a valuable contribution to clinical practice and should inform funder and policy decisions on rehabilitation for thumb carpometacarpal osteoarthritis.
--

REVIEWER	Holy, Chantal Johnson & Johnson Medical Devices, Real World Data Sciences
REVIEW RETURNED	23-May-2023

GENERAL COMMENTS	The authors have addressed all my comments. Below are some additional suggestions to help with clarity: 1) In the strengths and weaknesses: "technical or medical equipment" not included. I would specify which medical equipment (orthosis), as some assistive devices were reimbursed. 2) Last paragraph of the introduction: the cost-utility analysis of multimodal occupational therapy versus usual care ... (add the comparator: versus usual care) 3) Thank you for addressing the prior syntax issues, but on the question of "minimum wage": I would remove the article (a minimum wage). If you say "a minimum wage", it indicates that it was valued at a minimum of 18 Euro but could be valued at higher reimbursement too. If you say it was valued at minimum wage (without article), it means that the minimum legal wage in Norway of 18 Euro was used to value this cost item.
---

	Page 13, line 6-9 (redlined version). sorry I missed that in prior reviews, it's not immediately clear what " almost 1/3 of these costs were related to the predetermined assesment points" means? I would suggest clarifying. Discussion: first line - cost utility of intervention versus normal care (add comparator) Conclusion: ... "was cost-effective when provided in addition to usual care".
--	---